# Polygenic risk scores for cardiovascular diseases and type 2 diabetes

**Chi Kuen Wong**[1]*, **Enes Makalic**[2], **Gillian S. Dite**[1,2], **Lawrence Whiting**[1], **Nicholas M. Murphy**[1], **John L. Hopper**[2], **Richard Allman**[1,2]

1 Genetic Technologies Ltd., Fitzroy, Victoria, Australia, 2 Centre for Epidemiology and Biostatistics, The University of Melbourne, Melbourne, Victoria, Australia

* kevin.wong@gtglabs.com

**Data Availability Statement:** Access to the data used in this study can be obtained by applying directly to the UK Biobank at https://www.ukbiobank.ac.uk/register-apply/. The authors did not receive special access privileges to the data

## Abstract

Polygenic risk scores (PRSs) are a promising approach to accurately predict an individual's risk of developing disease. The area under the receiver operating characteristic curve (AUC) of PRSs in their population are often only reported for models that are adjusted for age and sex, which are known risk factors for the disease of interest and confound the association between the PRS and the disease. This makes comparison of PRS between studies difficult because the genetic effects cannot be disentangled from effects of age and sex (which have a high AUC without the PRS). In this study, we used data from the UK Biobank and applied the stacked clumping and thresholding method and a variation called maximum clumping and thresholding method to develop PRSs to predict coronary artery disease, hypertension, atrial fibrillation, stroke and type 2 diabetes. We created case-control training datasets in which age and sex were controlled by design. We also excluded prevalent cases to prevent biased estimation of disease risks. The maximum clumping and thresholding PRSs required many fewer single-nucleotide polymorphisms to achieve almost the same discriminatory ability as the stacked clumping and thresholding PRSs. Using the testing datasets, the AUCs for the maximum clumping and thresholding PRSs were 0.599 (95% confidence interval [CI]: 0.585, 0.613) for atrial fibrillation, 0.572 (95% CI: 0.560, 0.584) for coronary artery disease, 0.585 (95% CI: 0.564, 0.605) for type 2 diabetes, 0.559 (95% CI: 0.550, 0.569) for hypertension and 0.514 (95% CI: 0.494, 0.535) for stroke. By developing a PRS using a dataset in which age and sex are controlled by design, we have obtained true estimates of the discriminatory ability of the PRSs alone rather than estimates that include the effects of age and sex.

## Introduction

A polygenic risk score (PRS) is a single quantitative measure to capture the relationship between multiple genetic variants and a phenotype. In practice, it is usually calculated by the sum of risk allele counts of the single-nucleotide polymorphisms (SNPs) weighted by their effect sizes. A PRS can explain the relative risk of getting a particular disease compared to others with a different genotype.

that others would not have. Interested researchers will be able to access the data in the same manner by applying directly to the UK Biobank. The successful PRSs arising from this study have been deposited in PGS Catalog (PGS002773 – PGS002780). R code used in the analyses is available from the corresponding author for non-commercial purposes only.

**Funding:** The authors received no external funding for this work. CKW, GSD, LW, NMM and RA are employed by a commercial company, Genetic Technologies Limited, which provided support in the form of salaries but did not have any role in the study design, data collection and analysis, decision to publish, or preparation of the manuscript. The specific roles of all authors are articulated in the Author Contributions section.

**Competing interests:** I have read the journal's policy and the authors of this manuscript have the following competing interests: CKW, GSD, LW, NMM and RA are employees of Genetic Technologies Limited. Aspects of this manuscript are covered by Provisional Patent Application AU 2020903793, Methods of assessing risk of developing a disease. CKW, GSD, NMM and RA are named inventors on the patent application, which is assigned to Genetic Technologies Limited. This does not alter our adherence to PLOS ONE policies on sharing data and materials.

As the power of polygenic risk scores (PRSs) has substantially increased over the last few years due to more advanced computing technology and better computational algorithms, more studies have suggested that PRSs are capable of identifying clinically meaningful increases in risk prediction [1–4]. For example, Khera et al. [1] developed a PRS for coronary artery disease that identified 8% of individuals with greater than 3-fold increased risk, which is comparable to the increase in risk from monogenic mutations. The discriminatory power of these PRSs, as reported by the area under the receiver operating characteristic curve (AUC), have usually been quite high. For instance, the PRSs developed by Khera et al. [1] has an AUC of 0.81 for coronary artery disease and 0.77 for atrial fibrillation; the metaGRS developed by Inouye et al. [2] has an AUC of 0.79 for coronary artery disease, and Bolli et al. [5] developed a PRS that has an AUC of 0.81 for coronary artery disease.

However, the prediction models used in these studies adjust for age and sex, which are known risk factors for the disease of interest and confound the association between the PRS and the disease. The reference models in these studies, which often include age, sex and a few principal components, already have high AUCs. If these studies do not report AUCs separately for the reference model and the PRS, recognizing how much a PRS actually contributes to disease prediction is impossible.

In addition, comparison of AUCs obtained by including additional covariates between studies can be difficult because of differences in the age and sex distributions of the study sample. A disease with a non-linear association with age will have a different AUC in a study of younger people versus a study of older people. This is because the reference models (i.e., the age and sex models) have different AUCs. Not knowing the separate AUCs for the PRS and the reference model makes comparison difficult.

Another problem with some studies [1] that seek to develop PRS is the use of prevalent cases. This can lead to biased estimates of disease risks, known as the prevalence–incidence bias [6], because severe cases die before, or are too unwell for, study enrolment, leaving only the mild cases included in the analysis.

In this paper, we aim to develop PRSs for coronary artery disease, hypertension, atrial fibrillation, stroke and type 2 diabetes when the effects of age and sex are controlled by design. We deliberately created a matched case-control study in which we controlled for age and sex by sampling controls from the available data and we excluded prevalent cases to prevent potential mis-estimation of disease risks.

Predicting an individual's risk of developing disease can provide tremendous value in public health. It allows early intervention and less costly treatment by directing screening or other health resources to the patients who are at high risk.

## Materials and methods

### Ethics approval

The UK Biobank has Research Tissue Bank approval (REC #11/NW/0382) that covers analysis of data by approved researchers. All participants provided written informed consent to the UK Biobank before data collection began. This research has been conducted using the UK Biobank resource under Application Number 47401.

### Participants

We used genotyped data from the UK Biobank Axiom Array [7] to develop PRSs for five common diseases [8, 9]: coronary artery disease, hypertension, atrial fibrillation, stroke and type 2 diabetes. The UK Biobank conducted baseline assessment of over 500,000 participants aged 40–69 years from 2006 to 2010. We used the disease definitions described in the supplements

of Said et al. [8, 9]. Prevalent cases were excluded to prevent biased estimation of disease risks. For quality control, we removed variants with minor allele frequency less than 0.001, Hardy–Weinberg equilibrium p-value less than $10^{-5}$, and genotyping rate of at least 95%. For each disease, the cases were split into training (70%) and testing (30%) datasets. The training datasets were used to build a PRS for each disease and the predictive performances were evaluated on the testing datasets.

## Training dataset

To control for the effects of age and sex, we applied the following sampling strategy to create the training datasets. For each of the five diseases, we computed the quintiles of age using the cases and divided individuals into five age groups. For each of the five age groups, and for each gender separately (i.e., total number of groups is 10 for each disease), we sampled 5 controls for each case. If the number of controls was not enough to draw 5 controls per case for all groups, we drew 4 per case, and so on. We then randomly selected 70% of the case and controls sets to form the training dataset. By this sampling strategy, the case-control ratios were approximately the same across all groups, and therefore the individuals were age and sex matched.

## Testing dataset

For the testing dataset for each of the diseases, we used the remaining 30% of cases and randomly sampled 5 controls per case without matching for age and sex. Controls were drawn from the 30% of unaffected participants identified or the testing dataset. The sizes of the training and testing datasets for each disease are summarized in Table 1.

## Statistical analysis

We created our PRSs using a recently developed method called stacked clumping and thresholding (SCT) [10]. Applying clumping (or pruning) to control linkage disequilibrium followed by marginal p-value thresholding is a standard method for computing PRS [11]. This approach requires users to specify hyperparameters such as the size of clumping windows ($kb$), the correlation threshold ($r^2$) and the p-value significance threshold for clumped SNPs. In general, it is not straightforward how to choose these hyperparameters in practice. Usually, users apply default values for these hyperparameters; for example, the default option in Plink [12] uses $r^2 = 0.5$ for the correlation threshold, 250 $kb$ for the window size and $p = 0.01$ for the p-value threshold.

SCT is an advanced algorithm that is based on the standard clumping and thresholding method. The user selects a set of values for each of the hyperparameters, runs clumping and thresholding on each combination of those parameters and gives a PRS for each combination. These steps can be efficiently conducted using the R package bigsnpr [13]. The PRSs are then stacked using a penalized regression model. The outcome of this algorithm is a linear combination of PRSs, where each PRS is a linear combination of variants. Therefore, a single vector of variant effect sizes can be obtained in the final prediction model. Instead of stacking these

**Table 1. Sizes of training and testing datasets used in our study.**

| Disease | Training size (controls/cases) | Testing size (controls/cases) |
|---|---|---|
| Coronary artery disease | 38,217 (31,847 / 6,370) | 16,374 (13,645 / 2,729) |
| Hypertension | 51,273 (41,018 / 10,255) | 21,970 (17,576 / 4,394) |
| Atrial fibrillation | 28,725 (23,937 / 4,788) | 12,306 (10,255 / 2,051) |
| Stroke | 12,693 (10,577 / 2,116) | 5,439 (4,533 / 906) |
| Type 2 diabetes | 12,768 (10,640 / 2,128) | 5,472 (4,560 / 912) |

**Table 2. A grid of hyperparameters used in the SCT algorithm.**

| Hyperparameters | Values |
|---|---|
| Correlation threshold ($r^2$) | 0.01, 0.05, 0.1, 0.2, 0.5, 0.8, 0.95 |
| Base window sizes ($kb$) | 50, 100, 200, 500 |
| Significance threshold ($p$) | 50 evenly spaced thresholds |

These are the default values used in the R package bigsnpr. The algorithm runs clumping and thresholding on each combination of these parameters and combines the risk scores by a penalized regression. The window size is computed as the base windows size divided by the correlation threshold. The significance threshold is evenly spaced on a logarithmic scale.

PRSs, we could select the PRS with the best prediction, and this is referred to as the maxCT approach. In general, SCT would identify more genetic variants than maxCT.

We applied SCT and maxCT to create PRSs for each of the five diseases. We used the training datasets to create 1,400 risk scores for each chromosome using the default hyperparameters values provided by the R package bigsnpr (Table 2). For maxCT, we selected the risk score that maximized the AUC on the training datasets as the final PRS. For SCT, we stacked the 30,800 (1,400 × 22) risks scores from all 22 chromosomes using penalized logistic regression; the optimal stack weight was also estimated from the training sets.

To estimate the GWAS effect sizes of SNPs, we obtained summary statistics from large external GWAS. We removed ambiguous SNPs and variants with duplicated positions or refSNP cluster ID numbers, and only kept SNPs that appeared in both the UK Biobank data and the study from which we used summary statistics. These GWAS and the number of SNPs are summarized in Table 3. The second last column gives the number of SNPs in the original studies. The last column shows the number of SNPs that appeared in both the UK Biobank data and the original studies after removing ambiguous SNPs and other quality control.

After PRSs were created for each disease, we quantified their predictive power in the testing data by using the bigstatsr package in R to compute AUCs. No other covariates were included in the calculation of the AUCs. To assess the association of each of the PRS with the disease of interest in the testing data, we used logistic regression to estimate the odds ratio (OR) per standard deviation (SD) of the PRS. The SDs were calculated using the controls in the 30% testing dataset. In addition, we assessed the calibration performance by fitting a logistic regression with the disease status and the logit of the predicted probabilities given by our PRSs. A well-calibrated model should have an intercept close to 0 and a slope close to 1.

## Results

S1 Table shows, for each disease, the number of participants in the 70% age- and sex-matched training dataset and the number in the 30% unmatched testing dataset. Five controls were

**Table 3. External GWAS summary statistics used in our study.**

| Disease | GWAS study | # SNPs | # matched SNPs |
|---|---|---|---|
| Coronary artery disease | Nikpay et al. (2015) [14] | 9,455,778 | 506,432 |
| Hypertension | Zhu et al. (2019) [15] | 5,265,189 | 382,924 |
| Atrial fibrillation | Christophersen et al. (2017) [16] | 11,792,062 | 508,687 |
| Stroke | Malik et al. (2018) [17] | 8,255,860 | 513,802 |
| Type 2 diabetes | Scott et al. (2017) [18] | 12,056,346 | 536,788 |

able to be selected for almost all cases; three controls were not able to be matched for coronary artery disease, atrial fibrillation and stroke. For hypertension, four controls were drawn for each case and two controls were not able to be matche.

The distributions of the standardized SCT PRSs in the testing datasets are plotted in Fig 1. For all diseases except stroke, the PRSs for the cases had a greater mean and median than the controls. For example, for atrial fibrillation the PRS had a mean of 0.34 for the cases and −0.07 for the controls. Similarly, the PRS for coronary artery disease had a mean of 0.26 for the cases and −0.05 for the controls, and the PRS for type 2 diabetes had a mean of 0.28 for the cases and −0.06 for the controls. The PRS for stroke had similar mean for the cases and controls, 0.03 for the cases and −0.01 for the controls respectively, which shows the lack of discriminatory ability compared to other diseases.

The main results are summarized in Fig 2 and Table 4. The strongest predictive performance was found for the PRSs for atrial fibrillation followed by the PRSs for type 2 diabetes and coronary artery disease and then the PRSs for hypertension. The PRSs for stroke were unable to predict disease.

For each disease except stroke, the SCT PRS had a slightly higher AUC than the maxCT PRS but was based on many more SNPs: 820× more for atrial fibrillation, 370× more for coronary artery disease, 10× more for both type 2 diabetes and stroke, and 5× more for hypertension. For example, for atrial fibrillation the AUC increased from 0.599 (95% CI: 0.585, 0.613) for 265 SNPs in the maxCT PRS to an AUC of 0.613 (95% CI: 0.599, 0.626) for 216,837 SNPs in the SCT PRS. For hypertension, which had the largest number of maxCT SNPs, the AUC increased from 0.559 (95% CI: 0.550, 0.569) for 61,669 SNPs in the maxCT PRS to an AUC of 0.566 (95% CI: 0.556, 0.576) for 309,759 SNPs in the SCT PRS. The optimal hyperparameters for maxCT are reported in Table 5. These hyperparameters maximized the AUC in the training sets. These hyperparameters are the size of clumping

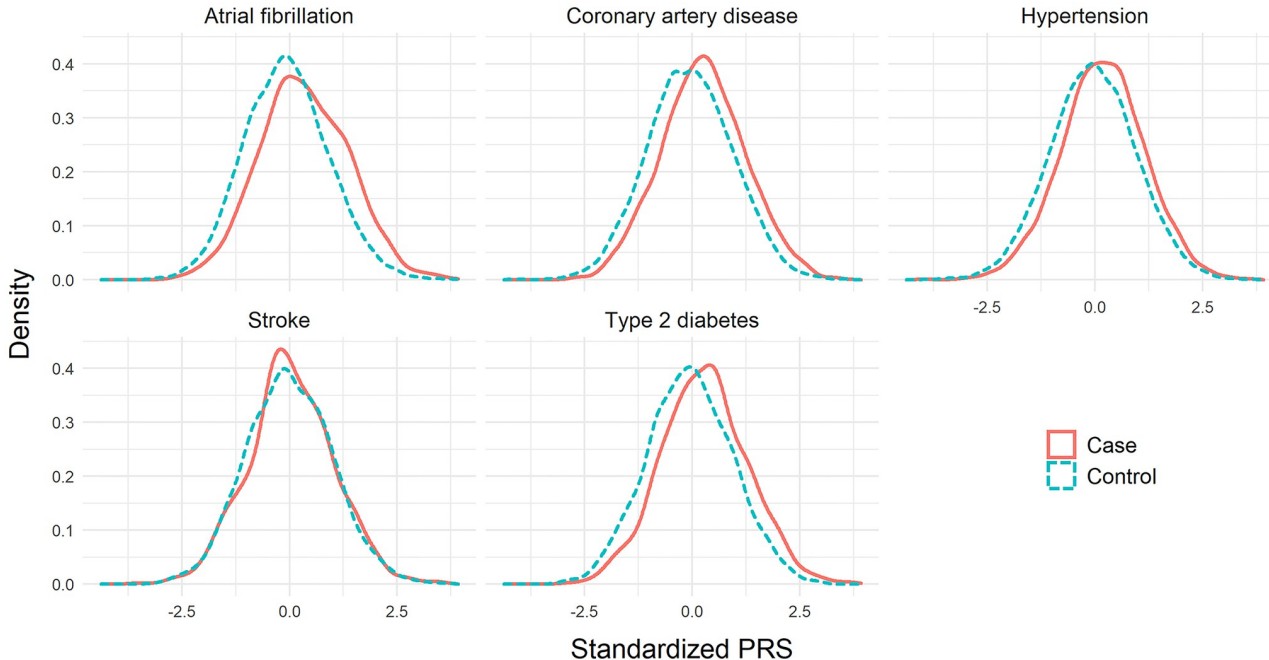

**Fig 1. Distribution of the standardized SCT PRSs (with mean 0 and standard deviation 1) for the cases and controls in five common diseases.**

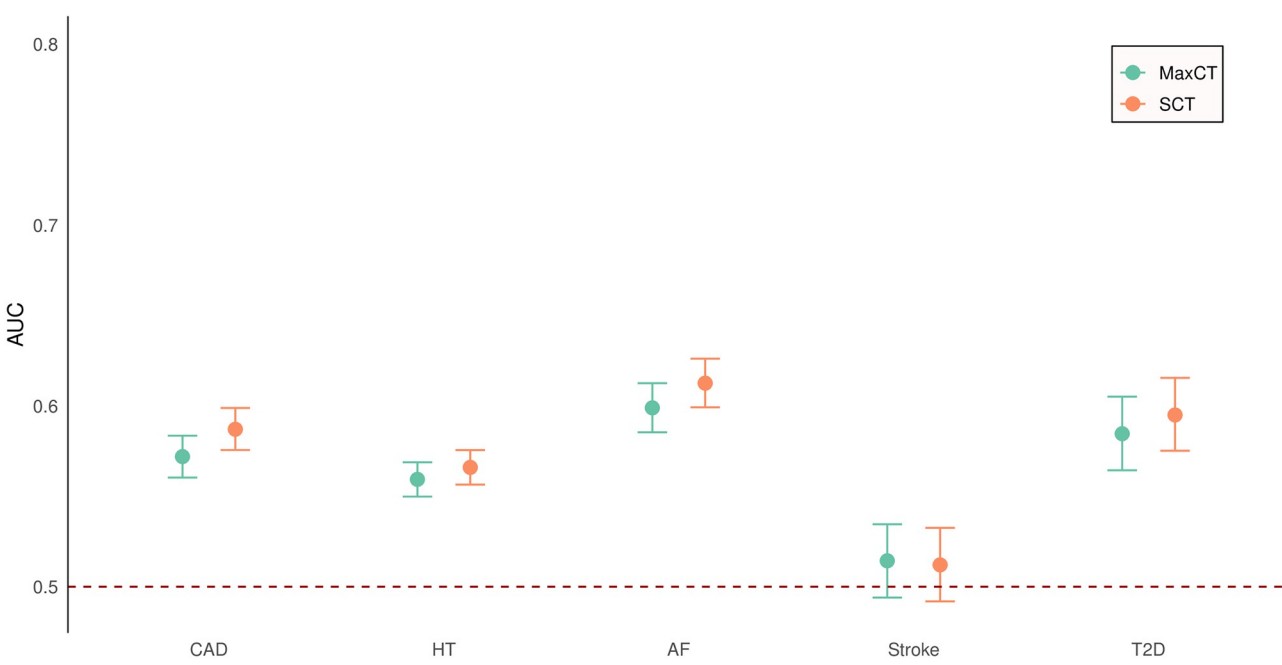

**Fig 2. Predictive performance of the PRSs generated by maxCT and SCT for five common diseases as measured by AUC.**

**Table 4. Predictive performance of the developed PRSs and the number of identified SNPs.**

| Disease | AUC (95% CI) | | Number of SNPs | |
|---|---|---|---|---|
| | **maxCT** | **SCT** | **maxCT** | **SCT** |
| Coronary artery disease | 0.572 (0.560, 0.584) | 0.587 (0.576, 0.599) | 1,059 | 390,782 |
| Hypertension | 0.559 (0.550, 0.569) | 0.566 (0.556, 0.576) | 61,669 | 309,759 |
| Atrial fibrillation | 0.599 (0.585, 0.613) | 0.613 (0.599, 0.626) | 265 | 216,837 |
| Stroke | 0.514 (0.494, 0.535) | 0.512 (0.492, 0.533) | 17,568 | 169,186 |
| Type 2 diabetes | 0.585 (0.564, 0.605) | 0.595 (0.575, 0.615) | 46,353 | 419,209 |

**Table 5. Optimal hyperparameters for maxCT.**

| Disease | $r^2$ | kb | p |
|---|---|---|---|
| Coronary artery disease | 0.50 | 1,000 | $1.23 \times 10^{-3}$ |
| Hypertension | 0.80 | 625 | $9.77 \times 10^{-2}$ |
| Atrial fibrillation | 0.95 | 52 | $5.75 \times 10^{-5}$ |
| Stroke | 0.95 | 105 | $2.82 \times 10^{-2}$ |
| Type 2 diabetes | 0.80 | 125 | $7.59 \times 10^{-2}$ |

windows (kb), the correlation threshold ($r^2$) and the p-value significance threshold for the clumped SNPs.

The coefficients of our calibration analysis showed that the prediction models were well calibrated for atrial fibrillation, coronary artery disease and type 2 diabetes. For the maxCT PRSs, the intercept and slope were found to be 0.02 (95% CI: −0.17, 0.22) and 1.02 (95% CI: 0.90, 1.14) for atrial fibrillation, −0.11 (95% CI: −0.31, 0.09) and 0.94 (95% CI: 0.81, 1.07) for

**Table 6. Odds ratio (and 95% confidence interval) per standard deviation for PRSs generated by maxCT and SCT.**

| Disease | maxCT | SCT |
|---|---|---|
| Coronary artery disease | 1.29 (1.24, 1.35) | 1.36 (1.31, 1.42) |
| Hypertension | 1.23 (1.18, 1.27) | 1.26 (1.22, 1.30) |
| Atrial fibrillation | 1.41 (1.35, 1.48) | 1.49 (1.42, 1.57) |
| Stroke | 1.05 (0.97, 1.13) | 1.04 (0.97, 1.12) |
| Type 2 diabetes | 1.35 (1.25, 1.45) | 1.41 (1.31, 1.51) |

**Table 7. Predictive performance of the SCT PRSs in the testing data, with and without including age and sex.**

| Disease | AUC (95% CI) | |
|---|---|---|
| | PRS only | PRS + sex +age |
| Coronary artery disease | 0.587 (0.576, 0.599) | 0.706 (0.696, 0.716) |
| Hypertension | 0.566 (0.556, 0.576) | 0.677 (0.669, 0.686) |
| Atrial fibrillation | 0.613 (0.599, 0.626) | 0.738 (0.728, 0.750) |
| Stroke | 0.512 (0.492, 0.533) | 0.668 (0.650, 0.686) |
| Type 2 diabetes | 0.595 (0.575, 0.615) | 0.638 (0.619, 0.657) |

coronary artery disease, and −0.13 (95% CI: −0.45, 0.17) and 0.91 (95% CI: 0.72, 1.11) for type 2 diabetes. There was no evidence to reject the null hypothesis that the intercept of the calibration curve is zero and the slope is one. However, different results were found for the other two diseases, the calibration was weak for hypertension (slope = 0.80, 95% CI: 0.69, 0.92) and poor for stroke (slope = 0.37, 95% CI: −0.27, 1.01).

The results of the logistic regression to estimate the odds ratio (OR) per standard deviation (SD) are summarized in Table 6. They are similar to what we observed in terms of AUC, with the SCT PRSs having slightly higher associations than the maxCT PRSs. The strongest performance was seen for the PRSs for atrial fibrillation, having an OR per SD of 1.49 (95% CI: 1.42, 1.57) for the SCT PRS and an OR per SD of 1.41 (95% CI: 1.35, 1.48) for the maxCT PRS. The OR per SD for type 2 diabetes and coronary artery disease were similar in magnitude and the OR per SD for hypertension was slightly lower. The PRSs for stroke were not associated with disease. Table 7 shows the comparison of the AUCs for the SCT PRSs with and without the inclusion of age and sex in the testing data.

For each disease, we selected summary statistics from the GWAS Catalog that are publicly available for download, discovered using mostly Caucasian populations with a large sample size, and not generated using the UK Biobank. Ideally, we would prefer to use summary statistics that used only incident cases (and also satisfied the other mentioned criteria) in order to match our study design but we cannot find such summary statistics. We would expect better performance if the summary statistics were commensurate with our study design. Note that for hypertension, the summary statistics were generated using the UK Biobank so the performance of the hypertension PRS could be overestimated.

## Discussion

In this study, we have addressed two important limitations of some other studies that have attempted to develop PRSs [1, 2, 5]. First, we have ensured that our PRS models do not include the effects of age and sex and represent the genetic effects alone.

To do this, we used a sampling strategy to create training datasets in which age and sex are controlled by design. We ensured that the ratio of the number of cases to the number of controls was the same across all age and sex groups in the training datasets. Therefore, the selection of SNPs and estimation of their ORs in the development stage of the PRSs cannot be affected by age and sex. Importantly, the AUCs for our PRSs in the testing datasets are due solely to genetic effects and are not inflated from the inclusion of age and sex in the model. For example, in the testing dataset for atrial fibrillation, the AUC for a base model with age and sex was 0.711, while the AUC for our PRS alone was 0.613 (see Table 7). If we present the AUC for PRS, age and sex–as other authors [1, 5] have done–it would be 0.738. This age- and sex-adjusted AUC has often been reported without also reporting an AUC for the PRS alone, making it difficult to understand the contribution of the PRS to disease prediction.

Second, in other studies, the inclusion of prevalent cases might lead to biased estimation of disease risks because severe or fatal cases do not have an opportunity to be included in the analysis. By excluding prevalent cases we have ensured that our disease risks are not mis-estimated. The vast size of the UK Biobank has meant that we have achieved large sample sizes using incident cases.

Our results suggest that the PRSs developed in this study have moderate discriminatory power for incident atrial fibrillation (AUC = 0.613), coronary artery disease (AUC = 0.587) and type 2 diabetes (AUC = 0.595). Our PRSs were not able to predict risk for stroke. It has been pointed out in a previous study [3] that PRS for stroke is less predictive than PRS for other common diseases because stroke is a more heterogeneous disease. Including more variants in a PRS can improve its predictive performance, even if most of the variants have very small effect sizes [5]. While this is consistent with our findings that the SCT PRSs have better performance than the maxCT PRSs, the cost and the practicality of implementing these PRSs into clinical practice should also be taken into consideration. Finding the balance between performance and practicality is crucial for a successful implementation. We found that the maxCT PRS could be a good candidate for this purpose because the number of SNPs is much more manageable (e.g., for atrial fibrillation, 265 SNPs for the maxCT PRS vs. 216,837 SNPs for the SCT PRS) without too much sacrifice of the prediction performance (e.g., AUC of 0.599 with maxCT vs 0.613 with SCT for atrial fibrillation). Simulation and real data analysis [10] has shown that maxCT outperforms the most widely used clumping and thresholding method.

One potential limitation of our study is that we used summary statistics from GWAS that were not matched for age and sex. This approach will potentially reduce the performance of the PRSs. We used external summary statistics rather than obtaining them using a hold-out set from the UK Biobank so that we could maximize the samples available for analysis. While the weights from the summary statistics are used in the maxCT PRSs, we selected SNPs using the training data, which is age- and sex- matched. For the SCT PRSs, the final weights of the SNPs are obtained by fitting a penalized regression using the training data.

The most common cardiovascular disease is coronary artery disease (CAD) which involves the reduction of blood flow to the heart muscle due to build-up of plaque (atherosclerosis) in the arteries of the heart. Clinical risk factors include high blood pressure, smoking, diabetes, lack of exercise, obesity, high blood cholesterol, poor diet, depression and excessive alcohol. Similarly Type 2 diabetes primarily occurs as a result of modifiable risk factors. Thus, accurate risk prediction for the development of these diseases allows early intervention, including educational resources to drive behavior modification, to the patients who are at high risk.

Established clinical risk prediction scores, for example, the Framingham risk scores, are designed for use in people aged over 30 years [19] and some studies [20] have shown that

individuals with high PRS had similar risk to the individuals with familial hypercholesterol-emia (a genetic disorder that increases the likelihood of coronary artery disease) although their levels of cholesterol and other traditional risk factors were normal. As a result, individuals at high genetic risk of coronary artery disease might not be receiving timely advice because of the limitation of the clinical risk tools. Because a PRS is based on germline DNA, it can potentially be used much earlier than conventional risk prediction tools. The early identification of individuals at increased genetic risk could lead to prevention strategies at earlier ages and significant savings in mortality and treatment costs.

## Conclusion

We developed PRSs and evaluated their predictive performances for coronary artery disease, hypertension, atrial fibrillation, stroke and type 2 diabetes. Using a sampling strategy, the effects of age and sex have been controlled by design and did not affect the development of the PRSs. The predictive performances were reported as their true AUCs, not AUCs that include the effects of age and sex. Our PRSs have moderate predictive power to predict incident coronary artery disease, atrial fibrillation and type 2 diabetes. Further study should be investigated to examine the clinical utility for PRS to improve risk predictions for these diseases.

## Supporting information

**S1 Table. Number, age quintile and sex of the cases and controls in the 70% training matched dataset for each of the diseases studied.**
(DOCX)

## Author Contributions

**Conceptualization:** Chi Kuen Wong, Enes Makalic, Gillian S. Dite, John L. Hopper, Richard Allman.

**Data curation:** Gillian S. Dite, Lawrence Whiting, Nicholas M. Murphy.

**Formal analysis:** Chi Kuen Wong.

**Investigation:** Chi Kuen Wong, Enes Makalic, Gillian S. Dite, Lawrence Whiting, Nicholas M. Murphy.

**Methodology:** Chi Kuen Wong, Enes Makalic, Gillian S. Dite, John L. Hopper, Richard Allman.

**Project administration:** Richard Allman.

**Software:** Chi Kuen Wong.

**Writing – original draft:** Chi Kuen Wong.

**Writing – review & editing:** Chi Kuen Wong, Gillian S. Dite, John L. Hopper, Richard Allman.

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
