## [Decision Letter · Decision Letter 0]

7 Sep 2022

PONE-D-22-03370Polygenic risk scores for cardiovascular diseases and type 2 diabetesPLOS ONE

Dear Dr. Wong,

Thank you for submitting your manuscript to PLOS ONE. After careful consideration, we feel that it has merit but does not fully meet PLOS ONE’s publication criteria as it currently stands. Therefore, we invite you to submit a revised version of the manuscript that addresses the points raised during the review process.

The manuscript has been evaluated by two reviewers that both raised major concerns with the current version. Specifically, they are concerned about the reporting and inclusion of the sex and age variables and request clarifications on the statistical models that have been used.Their full reviewers are attached below, could you please revise your manuscript to carefully address all their concerns?

We look forward to receiving your revised manuscript.

Kind regards,

Thomas Tischer

Staff Editor

PLOS ONE

Journal Requirements:

2. Thank you for providing the following Funding Statement: 

“I have read the journal's policy and the authors of this manuscript have the following competing interests: CKW, GSD, LW, NMM and RA are employees of Genetic Technologies Limited. Aspects of this manuscript are covered by Provisional Patent Application AU 2020903793, Methods of assessing risk of developing a disease. Chi Kuen Wong, Gillian Dite, Nicholas Murphy and Richard Allman are named inventors on the patent application, which is assigned to Genetic Technologies Limited.”

We note that one or more of the authors is affiliated with the funding organization, indicating the funder may have had some role in the design, data collection, analysis or preparation of your manuscript for publication; in other words, the funder played an indirect role through the participation of the co-authors.

If the funding organization did not play a role in the study design, data collection and analysis, decision to publish, or preparation of the manuscript and only provided financial support in the form of authors' salaries and/or research materials, please review your statements relating to the author contributions, and ensure you have specifically and accurately indicated the role(s) that these authors had in your study in the Author Contributions section of the online submission form. Please make any necessary amendments directly within this section of the online submission form.  Please also update your Funding Statement to include the following statement: “The funder provided support in the form of salaries for authors [insert relevant initials], but did not have any additional role in the study design, data collection and analysis, decision to publish, or preparation of the manuscript. The specific roles of these authors are articulated in the ‘author contributions’ section.”

If the funding organization did have an additional role, please state and explain that role within your Funding Statement.

Please also provide an updated Competing Interests Statement declaring this commercial affiliation along with any other relevant declarations relating to employment, consultancy, patents, products in development, or marketed products, etc. 

Reviewers' comments:

Reviewer's Responses to Questions

**Comments to the Author**

1. Is the manuscript technically sound, and do the data support the conclusions?

Reviewer #1: Yes

Reviewer #2: Partly

2. Has the statistical analysis been performed appropriately and rigorously? 

Reviewer #1: Yes

Reviewer #2: Yes

3. Have the authors made all data underlying the findings in their manuscript fully available?

Reviewer #1: No

Reviewer #2: No

4. Is the manuscript presented in an intelligible fashion and written in standard English?

Reviewer #1: Yes

Reviewer #2: Yes

5. Review Comments to the Author

Reviewer #1: In this study, Wong et al. derive polygenic risk scores (PRS) for incident cases of several cardiovascular diseases and type 2 diabetes using a clumping and thresholding approach. They use a subset (70%) of the UKBB that they divide into age and sex matched cases and controls as a trainining set to optimize hyperparameters for their clumping and thresholding approach for each phenotype respectively. They then test the predictive performance of their PRS on the remaining 30% test set and state, that their sampling strategy controlled for the effects of age and sex in the development of the PRS.

This study has several major issues in the design of the experiment, as well as in the reporting. I think the authors need to address all of these concerns to show that their results reflect the claims they are making. I have listed alls of my comments regarding the manuscript below:

General major points

1) The authors mix a couple of different arguments in their abstract and introduction about how age and sex are problematic in the construction of PRS and how this apparently affects AUC reporting of predictive models. I agree with the authors, that predictive performance of PRS is often reported including age and sex as covariates in the logistic regression models and that authors should report base models without PRS as well, to properly assess how much the PRS adds ot the performance. However, the argument that age and sex are included in the PRS models themselves is questionable. The authors should clarify why they think that matching cases and controls for age and sex in the clumping and thresholding step of the PRS construction accounts for age and sex differences for traits, since the clumping and thresholding approach will select candidate variants from pre-calculated GWAS results based p-value thresholds and LD-structure in the training set. If the authors believe that age and sex are associated with the traits that they are investigating, shouldn’t they rather perform GWAS stratified by age and sex to recalibrate effect sizes and association statistics? This way, the variants that will be selected for each respective strata during clumping and thresholding would actually reflect the differences the authors want to highlight. An example of sex stratified PRS generation can be found in PMID: 35873490 (https://pubmed.ncbi.nlm.nih.gov/35873490/).

2) Related to the issue of how the authors attempt to control for age and sex in the construction of their PRS, they fail to show, that by not controlling for these covariates, models are actually over-or underperforming. The authors should generate PRS using their same approach without matching cases-and controls and report the predictive performance of these models to show whether they actually differ across sex and age groups compared to their age-and sex matched PRS.

3) The authors removed prevalent disease cases from their analysis. They do highlight that this is to prevent biased estimation of disease risk, which is an important point to make. However, they fail to also highlight that by using prevalent disease based GWAS summary statistics in the selection of their PRS variants, their PRS might also be underperforming, as variants selected for PRS might be biased towards prevalent disease.

4) In the methods section, the authors write that they computed AUC values to assess predictive performance of their PRS, but fail to provide details about which computational packages they used (if they used any) to do this. In the results section, they refer to using logistic regression to calculate the Odds ratio (OR) per standard deviation for Table 3. Were the probabilities from the logistic regression models used for calculation of AUC and did the authors include any covariates in the logistic regression models (PCs, smoking, statin usage, etc?). The authors need to add the details about logistic regression models for OR per SD calculation to the methods section. Currently, what they write in the results section does not match what they report in the methods section for OR per SD calculation.

Minor points

One controversial point the authors raise is whether variant imputation is too computationally and analytically intense to be generally applied to PRS calculation. They do cite a study that investigated the effect of different imputation algorithms on PRS performance (Ref 8 in the manuscript : https://genomemedicine.biomedcentral.com/articles/10.1186/s13073-020-00801-x). However, while this study found that imputation can introduce variability on the individual level, it generally does not cause problems in interpretation of the PRS. As many easy-to-use public tools such as the Michigan Imputation server (https://imputationserver.sph.umich.edu/index.html) are providing low-cost, fast imputation to large reference panels and whole-genome sequencing is becoming cheaper and cheaper, the argument that imputation and large PRS panels are unfeasible becomes less valid. The Michigan imputation server has recently even incorporated PRS calculation (from PGS catalog) as part of the imputation process, making it even easier and faster to obtain PRS for datasets.

Data reporting

The authors fail to mention availability of their score for the public to reproduce their results in independent cohorts. They don’t provide any reference to a data or any other repository where their PRS variants and effect sizes can be found. I suggest the authors submit their PRS to https://www.pgscatalog.org/, a resource that has been created for exactly the purpose of making PRS reporting more reproducible.

Reviewer #2: The authors describe that AUC reflects the predictive accuracy of not only the polygenic score but also important risk factors, namely age and sex. The authors use sex- and age-matched control samples to estimate the AUC of polygenic score alone.

I have some major concerns. Firstly, it is not very clear to me what the aim of this work is and what contribution to the field the authors are trying make.

(1) If the aim is to show that the commonly used AUC estimates are ‘inflated’ due to the effects of age and sex, what is missing in the manuscript is a proper comparison between AUC estimates using the same polygenic scores in case control samples with and without matching the distributions of sex and age. It was only briefly mentioned in the discussion section (line 139). It would always be helpful to quantify the observations.

(2) If the aim is to develop polygenic scores that improve risk stratification for the tested traits (as mentioned on line 65 in the introduction), the authors did not compare their PRS with previously published scores, some of which used more sophisticated methods such as Bayesian based methods (e.g. LDpred, PRSCS, and SBayesR). These methods are believed to outperform C+T based methods. It can probably also address the issue that the stroke score was not predictive, by using the PRS developed by Neumann et al. (available at the PGS Catalog; https://www.ahajournals.org/doi/10.1161/STROKEAHA.120.033670#d6462236e293)or from an older study that the authors themselves cited (reference 3).

Moreover, I don’t agree with the authors that adjusting or considering age and sex in the model when reporting the ‘inflated’ AUC of a polygenic score is an issue. We compare AUC basically in the following two scenarios. (1) We compare different polygenic scores or scores generated using different parameters in the same testing sample. Age, sex and other baseline characteristics have the same effects on the goodness-of-fit metrics and better scores always have higher AUC. (2) We compare scores that are validated in different cohorts. It is more complicated and there are always other cohort-specific factors (such as different healthcare systems, phenotype definitions, fine-scale population stratification, different socioeconomic status) which may contribute to the estimation of PRS accuracy. Even the case control samples within each cohort have matched age and sex distributions, differences in the distributions between cohorts (e.g. one cohort might have many young cases while the other cohort recruits more older cases) could still result in AUC estimates that are not comparable. Also, power is reduced when selecting controls that match with cases due to the smaller sample size. There are other approaches to accounting differences in age and sex between cases and controls, which have been commonly used already, such as the incremental R2 (AUC or pseudo-R2 for binary traits; https://doi.org/10.1038/s41596-020-0353-1) which quantifies the increase in variance explained with the addition of the PRS to the baseline model.

Also, the authors discussed a lot about the disadvantages of using imputed versus genotyped array data, which doesn’t really fit in the manuscript. The authors did not do any analysis comparing the accuracy, individual-level variability, or costs versus benefits between PRS with a small number of genetic variants using array data and PRS with many more genetic variants using imputed data. This is out of the scope of the work and the discussion on this topic probably needs to be shortened. The justification of using array data can be briefly mentioned in the beginning of the results section when the study samples and analysis methods are introduced, or in the methods section.

More specific comments:

Results section on page 5: it would be great to briefly introduce the study cohort first. More specifically, the effort of matching controls with cases in terms of age and sex should be mentioned, as well as the sample sizes. It would be great to move table 4 here.

Line 76: figure 2 shows the distributions, not figure 1.

Page 7: I’m not very sure whether the comparison of SCT and maxCT scores is necessary here. It has been clearly established in the original paper (citation 7) that the SCT scores show better performance than maxCT. It is not surprising either that the SCT scores contain more genetic variants given that they are combinations of multiple C+T scores.

Line 151: it would be more accurate to say that “our results suggest that PRSs that were developed in this study have xxx”. The conclusion may not be generalisable to other PRSs.

Line 168-169: it should be made clearly in the beginning of the results section that the genotyped data were used.

Page 10-11: like I said previously, there are a lot of discussions about imputed data, which is not the focus of this manuscript which currently does not have any analysis relevant to imputed vs genotyped data.

The authors should also consider that genotyping only a small number of SNPs that are used in the maxCT scores is not flexible. There will be larger GWAS and more accurate PRS for the tested diseases as well as new phenotypes in the future, and different genetic markers will be needed. In the long term, using commercial array chips + imputation strategy is probably more cost effective, as we can reuse the high coverage data for better scores and scores for many other traits when available.

Line 171-172: what does the ‘validation process’ mean here?

Line 185-186: please provide reference to this claim which I find hard to believe. There are standard ways to perform QC of array data and imputation can be done easily and freely using online servers.

6. PLOS authors have the option to publish the peer review history of their article (what does this mean?). If published, this will include your full peer review and any attached files.

Reviewer #1: No

Reviewer #2: No

---

## [Author Response · Author response to Decision Letter 0]

3 Oct 2022

Please see the attached file "Response to Reviewers"

---

## [Decision Letter · Decision Letter 1]

15 Nov 2022

PONE-D-22-03370R1Polygenic risk scores for cardiovascular diseases and type 2 diabetesPLOS ONE

Dear Dr. Wong,

Thank you for submitting your manuscript to PLOS ONE. After careful consideration, we feel that it has merit but does not fully meet PLOS ONE’s publication criteria as it currently stands. Therefore, we invite you to submit a revised version of the manuscript that addresses the points raised during the review process.

Indeed, although the authors responded to most of the comments raised by the reviewers, this editor and the reviewers believe that it is important to show that the PRS constructed using cases and matched controls for covariates such as age and sex is actually better than simply adjusting for them when calculating predictive performance.

We look forward to receiving your revised manuscript.

Kind regards,

Gualtiero I. Colombo, M.D., Ph.D.

Academic Editor

PLOS ONE

Journal Requirements:

Reviewers' comments:

Reviewer's Responses to Questions

**Comments to the Author**

1. If the authors have adequately addressed your comments raised in a previous round of review and you feel that this manuscript is now acceptable for publication, you may indicate that here to bypass the “Comments to the Author” section, enter your conflict of interest statement in the “Confidential to Editor” section, and submit your "Accept" recommendation.

Reviewer #1: All comments have been addressed

Reviewer #2: (No Response)

2. Is the manuscript technically sound, and do the data support the conclusions?

Reviewer #1: Yes

Reviewer #2: Yes

3. Has the statistical analysis been performed appropriately and rigorously? 

Reviewer #1: Yes

Reviewer #2: Yes

4. Have the authors made all data underlying the findings in their manuscript fully available?

Reviewer #1: Yes

Reviewer #2: Yes

5. Is the manuscript presented in an intelligible fashion and written in standard English?

Reviewer #1: Yes

Reviewer #2: Yes

6. Review Comments to the Author

Reviewer #1: The authors have adequately addressed all of my comments to their original manuscript submission. The scores submitted to the PGS Catalog need to be made public once the manuscript is accepted in a publication journal.

Reviewer #2: The authors have addressed most of my comments and the revised manuscript has been greatly improved. I still have one last comment on authors’ response to 5.2 and 5.7. It is important to establish and quantify the issue of over-estimation, as the authors claimed, of PRS accuracy measured in AUC when including age and sex. I still think that it would be great to analyse all traits in addition to atrial fibrillation (line 241-244), and add the “over-estimated” AUC somewhere in a main table. I find it interesting and helpful (and perhaps some other readers too) to know how much the AUC metrics reported in other studies are overestimated due to the effects of covariates such as age and sex.

7. PLOS authors have the option to publish the peer review history of their article (what does this mean?). If published, this will include your full peer review and any attached files.

Reviewer #1: No

Reviewer #2: No

---

## [Author Response · Author response to Decision Letter 1]

21 Nov 2022

Reviewer #1: The authors have adequately addressed all of my comments to their original manuscript submission. The scores submitted to the PGS Catalog need to be made public once the manuscript is accepted in a publication journal.

We will ensure that the PRSs submitted to the PGS Catalog are made public when the paper is accepted for publication.

Reviewer #2: The authors have addressed most of my comments and the revised manuscript has been greatly improved. I still have one last comment on authors’ response to 5.2 and 5.7. It is important to establish and quantify the issue of over-estimation, as the authors claimed, of PRS accuracy measured in AUC when including age and sex. I still think that it would be great to analyse all traits in addition to atrial fibrillation (line 241-244), and add the “over-estimated” AUC somewhere in a main table. I find it interesting and helpful (and perhaps some other readers too) to know how much the AUC metrics reported in other studies are overestimated due to the effects of covariates such as age and sex.

In additional to atrial fibrillation, we have now added Table 7 to show the inflation of AUCs when age and sex are included for all five diseases.

---

## [Editor Report · Decision Letter 2]

23 Nov 2022

Polygenic risk scores for cardiovascular diseases and type 2 diabetes

PONE-D-22-03370R2

Dear Dr. Wong,

We’re pleased to inform you that your manuscript has been judged scientifically suitable for publication and will be formally accepted for publication once it meets all outstanding technical requirements.

Kind regards,

Gualtiero I. Colombo, M.D., Ph.D.

Academic Editor

PLOS ONE
---

## [Editor Report · Acceptance letter]

25 Nov 2022

PONE-D-22-03370R2 

Polygenic risk scores for cardiovascular diseases and type 2 diabetes 

Dear Dr. Wong:

I'm pleased to inform you that your manuscript has been deemed suitable for publication in PLOS ONE. Congratulations! Your manuscript is now with our production department. 

Kind regards, 

on behalf of

Dr. Gualtiero I. Colombo 

Academic Editor

PLOS ONE